# School Students’ Concerns and Support after One Year of COVID-19 in Austria: A Qualitative Study Using Content Analysis

**DOI:** 10.3390/healthcare10071334

**Published:** 2022-07-18

**Authors:** Andrea Jesser, Yvonne Schaffler, Afsaneh Gächter, Rachel Dale, Elke Humer, Christoph Pieh

**Affiliations:** Department for Psychosomatic Medicine and Psychotherapy, University for Continuing Education Krems, 3500 Krems, Austria; yvonne.schaffler@donau-uni.ac.at (Y.S.); afsaneh.gächter@donau-uni.ac.at (A.G.); rachel.dale@donau-uni.ac.at (R.D.); elke.humer@donau-uni.ac.at (E.H.); christoph.pieh@donau-uni.ac.at (C.P.)

**Keywords:** adolescents, COVID-19, youth mental health, qualitative research

## Abstract

Adolescents suffer severely from the psychological consequences of the COVID-19 pandemic. Using qualitative content analysis, this study examined open-ended responses to a survey on the mental health of school students in Austria in February 2021. A representative sample (*n* = 214) was drawn from a total survey sample of 3052 adolescents aged 14–20. The analysis revealed several areas of concern, including school-related concerns, concerns about restrictions, self-related concerns, and interpersonal problems. School-related concerns associated with distance learning were mentioned most frequently. Compared with research conducted at the beginning of the pandemic, it appeared that concerns about educational and professional futures increased. The analysis also indicated young people’s most important sources of support, such as social contacts, recreational activities, attitudes and abilities, distraction, and escape. Of concern is the proportion of young people citing maladaptive coping strategies and the reluctance to seek professional support. Ideas for practice-oriented measures were developed from the study results, such as embedding youth-led peer interventions in traditional mental health services.

## 1. Introduction

The COVID-19 pandemic has dominated our lives for more than two years now. Governments worldwide have taken measures to contain the spread of the virus. In Austria, the first nationwide lockdown was decreed on 16 March 2020 and brought massive restrictions to public life. People were no longer allowed to enter public spaces except to meet necessary basic needs, to care for people in need of support, or for professional purposes. Schools and universities were closed. Private gatherings were prohibited, and face masks were mandatory in indoor public spaces. These measures were first gradually relaxed and then lifted entirely on 1 May 2020. Schools also reopened in mid-May. The measures were tightened again in autumn 2020 when the second wave of infection began to spread. As of November, higher secondary school students were again taught via distance learning. A second lockdown from 17 November to 6 December and a third lockdown from 26 December to 7 February 2021 came into place. During this time, primary and secondary school students returned to distance learning as well.

The psychological consequences of the pandemic give cause for concern [1,2,3]. Young people are particularly affected by the impact of the pandemic on mental health [4,5,6,7,8]. In February 2021, an Austrian study of 3052 adolescents aged 14 to 20 showed clinically relevant depressive symptoms in 55% of young people, anxiety symptoms in 47%, and sleep disorder symptoms in 23%. Suicidal thoughts were reported by 37% [9,10].

Studies on adolescent mental health during the COVID-19 pandemic are predominantly quantitative. There is considerably less research using qualitative methods to explore COVID-19-related mental health impacts [11]. Qualitative approaches may help to gain a better understanding of young people’s experience of living in times of a pandemic and learn about their concerns and sources of support from their perspective.

Existing qualitative studies have applied different methodological approaches to data collection and analysis. However, there are recurring themes found in most studies but with different emphases. Focusing on the challenges young people were experiencing at the onset of the pandemic, Scott et al. [12] questioned a sample of 719 US students aged between 14 and 19. Answers to an open-ended question were subjected to qualitative content analysis and the results indicated that respondents perceived academic issues as their biggest challenge, with 23.7% mentioning difficulties with distance learning, workload, preparing for standardized tests, focus, work ethic, and productivity. Other frequent categories were mental (14.8%) and physical (13.2%) health, and another 11.4% reported missing friends or friend conflict.

In general, these “challenges” are reflected in the findings of other studies, albeit in a different order, which could be due either to the methodological design of the studies or to the different course of the pandemic in other countries. A similar study of Portuguese adolescents highlighted the impact of the pandemic on social life [13]. Respondents reported suffering from the loss of social contacts, social competencies, and relevant life moments, such as going to their graduation ceremony. The young people found that their days were monotonous and boring and found themselves to be less productive without their usual routines. They reported more physical symptoms, such as headaches and muscle pain, that were associated with a less active life and more psychological symptoms, such as depression, anxiety, and loneliness. The importance of facing the pandemic positively, performing pleasurable activities, regular communication with family and friends, and establishing routines were mentioned as helpful coping mechanisms. The themes of this study were also found in an Irish study by O’Sullivan et al. [14], who collected additional perspectives on adolescents’ mental health by interviewing families.

A large Italian study involving 2758 young people showed yet another facet [15]. An emergent themes analysis revealed that loss of autonomy and anguish related to death and loss were the most frequent topics in young people’s narratives about recent adverse autobiographical events. This was followed by mentions of the new life routine of wearing masks and performing social distancing. While the loss of autonomy was also addressed in other studies, anguish related to death and loss was a central theme only in this study. The frequent occurrence of death themes might have been because Italy was the first European country that was hit particularly hard by the pandemic and suffered many casualties in the initial phase during spring 2020. The media coverage of COVID-19 in Italy was dominated by the themes of death and loss. However, the authors also demonstrated that adolescents could transform the COVID-19 experience into an opportunity for personal growth by using the lockdown as a chance for self-discovery. They found joy in sharing experiences with their peers at a distance and rediscovered family as a source of support. Thus, Fioretti et al. concluded rather optimistically that the COVID-19 pandemic could be both a disruptive and an empowering experience in young people’s lives.

The studies reviewed so far concentrated on reporting the negative and positive experiences of young people. Postigo-Zegarra et al. [16] went a step further by identifying different types of adaptation to the pandemic situation. They performed a discourse analysis of answers to open-ended survey questions about coping mechanisms and negative experiences during the first domestic lockdown in Spain. Adolescents who “adapted well” did so mainly by keeping busy with their academic and leisure activities, either alone or with friends. Technology was crucial for almost all the activities that helped them to stay active. They pointed out opportunities to do things that they usually could not. Adolescents with “moderate adjustment” reported feeling bored and tired. They mentioned some positive aspects of the situation but also mourned a loss of freedom. Adolescents who “failed in the adaptation process” did not perceive any positive or adaptive aspects of the pandemic situation. Their narratives were dominated by an experience of loss of routines, social contact, autonomy, freedom, and security. They described feeling socially isolated, locked in, and afraid of being infected with the virus. In addition, they felt overburdened by schoolwork.

While most qualitative studies focused on the initial phase of the pandemic in spring 2020, researchers from Norway conducted a focus group study with 15–19-year-old students in November and December 2020 after most students had faced a semester of distance learning, a return to the classroom at the start of the new school year, and a return to distance learning when the pandemic reached its second peak in October and November 2020 [17]. Qualitative content analysis revealed that the students’ well-being depended on the magnitude of restrictions. At this time, young people suffered the most from not seeing their friends and extended family. Distance learning and home confinement caused boredom and despair and the feeling of having missed opportunities to engage in meaningful life events. Furthermore, the interviewees expressed worries about their learning outcomes and prospects, as distance learning was not considered as effective as classroom teaching.

More up-to-date qualitative research is needed to shed light on the experience of young people as the pandemic progresses. Our study aimed at filling this gap, exploring young people’s concerns and sources of support after one year of COVID-19 in Austria in February 2021, when the government had just lifted the third lockdown and students were allowed to return to school. One objective of the study was to explore the underlying reasons for the high psychological burden on young people at the time of the survey: what do they describe as stressful? Second, we wanted to find out what helped them deal with stressful situations in order to make suggestions for interventions and preventive measures that build on the young people’s resources.

## 2. Materials and Methods

### 2.1. Research Design

We conducted a cross-sectional online survey of 14- to 20-year-old adolescents from 3 February to 28 February 2021. The current study was part of a larger study examining mental health in adolescents after the third COVID-19 lockdown in Austria and an extended period of distance learning. We received approval from the ethics committee and the data protection officer of the University for Continuing Education Krems (protocol code EK GZ 41/2018–2021).

The study was carried out using Research Electronic Data Capture (REDCap) [18,19], which is hosted at the servers of the University for Continuing Education Krems, Austria. The survey comprised 67 items measuring well-being (WHO-5), depression (PHQ-9), anxiety (GAD-7), sleep quality (ISI), stress (PSS-10), and disordered eating (EAT-8) [9,10,11,12,13,14,15,16,17,18,19]. To give respondents the opportunity to report in their own words and without a predefined set of answers regarding what concerned or helped them most in the current situation, the survey contained two open-ended questions: (Question 1) What currently gives you most cause for concern? (Question 2) What currently provides you with the most support? The original questions and answers were phrased in German. Responses ranged from single-word answers to entire paragraphs. Quantitative results were already published [9,10]; results from the qualitative analysis are presented in this paper.

### 2.2. Restrictions on Schooling in Austria

At the time of the survey, students had faced repeated and prolonged periods of distance learning. The new school year had started in September 2020. Since the beginning of November, higher secondary school students had been taught via distance learning. However, a shift system was implemented on 7 December, allowing them to return to school in alternating groups. This meant, for example, that half of the class was at school on Monday, Tuesday, and Wednesday, while the other half was taught via distance learning. From Thursday until Wednesday of the following week, the groups changed, and so on. In late November, the Austrian Minister of Education decided to postpone the school-leaving examination to give the school-leaving classes more time to prepare. In addition, the examination content was reduced by one-third, as the distance learning conditions were not considered comparable to classroom teaching [20].

### 2.3. Sampling Frame

The Austrian Federal Ministry of Education, Science and Research supported the study by informing all secondary schools in Austria about the project and asking the principals to forward the link to the survey to their students. Participation was voluntary without incentives. Participants had to agree to the data protection declaration to start the survey (electronic informed consent) and confirm they were aged 14 or older. The principles outlined in the Declaration of Helsinki were followed.

The total sample of N = 3052 was representative of Austria by region. Participants were on average M = 16.47 (SD = 1.44) years old; 70.1% were female, 28.1% male, and 1.8% non-binary; 16.6% had a migration background. A total of 2578 (84.5%) answered at least one of the two open-ended questions, while 2449 (80.2%) answered both.

Based on data from Statistics Austria [21], we drew a representative sample according to gender and migration background with N = 30 as the minimum in one category for the qualitative analysis. We chose these variables because previous research indicated that gender and migration background significantly impact adolescents’ mental health [22,23,24]. Dale et al. [25] showed that apprentices have worse mental health than high school students. For this reason, we excluded respondents who reported being apprentices, not attending a school at all, or doing something else, thereby focusing on young people attending compulsory school or some form of secondary school. Moreover, we only included respondents in the sample who answered both open-ended questions, regardless of the length of their comments.

### 2.4. Data Analysis

Data derived from open-ended questions were subjected to a conventional qualitative content analysis approach [26]. Two coders individually read through all the comments, inductively defining categories in the process. Subsequently, they compared their lists of categories and discussed the categories they differed on. Category definitions were added to the categories in this process and categories were assigned to thematic clusters.

In the next step, the two coders independently coded the whole data set with the list of categories. We used Atlas.ti [27] to make our coding process more transparent and enhance reliability [28]. To ensure consensual use of the categories, we assessed the agreement of how the two coders coded the data set [29]. At this stage in the coding process, the percentage agreement for “Support” was 80.4% inter-coder agreement using Krippendorff c-α-binary = 0.703. The percentage agreement for “Concerns” was 76.3% inter-coder agreement using Krippendorff c-α-binary = 0.788.

The principal investigator then checked the quotations where the coders disagreed and tried to solve conflicts. In most cases, this was easy because one coder had simply missed something or had selected the wrong category from the list. In other cases, the coding rules proved to be imprecise and had to be extended. After these corrections, the percentage agreement for “Support” was 96.4% inter-coder agreement using Krippendorff c-α-binary = 1.0. The percentage agreement for “Concerns” was 96.2% inter-coder agreement using Krippendorff c-α-binary = 1.0. This left 8 quotations (“Support”) and 11 quotations (“Concerns”) where the coders disagreed. These quotations were discussed within the research team [30] and a solution was found for each quotation separately.

In sum, the analysis of “Concerns” resulted in 9 categories and 28 sub-categories, and the analysis of “Support” resulted in 13 categories and 27 sub-categories. The two category systems are included in the Appendix A (Appendix A).

## 3. Results

### 3.1. Sample Description

Our final sample included 214 participants; 50% (N = 107) were female (compared with 49% female participants in the general population of 15–29-year-olds in Austria), 50% were male. A total of 28% of both female and male participants (N = 30) had a migration background (the proportion of people with a migration background was almost the same for boys and girls in this age group). Respondents were on average M = 16.43 (SD = 1.39) years old. A total of 1.4% (N = 3) of them attended middle school, 1.4% (N = 3) attended polytechnical school, 2.3% (N = 5) attended a school for intermediate vocational education, 45.8% (N = 98) attended a college for higher vocational education, and 49.1% (N = 105) attended an academic secondary school.

### 3.2. Greatest Current Concerns

In the following, we report the results for question 1: What currently gives you the most cause for concern? The results for question 1 are summarized in Figure 1 and are now described in more detail. The subsections in this section correspond to the main categories of the analysis. Section 3.2.1 describes the largest category “school-related concerns”, Section 3.2.2 summarizes statements related to the restrictions caused by the measures to combat the virus, Section 3.2.3 focuses on the self-related concerns reported by respondents, and Section 3.2.4 gives an overview of the remaining smaller categories.

#### 3.2.1. School-Related Concerns (63.1%)

N = 135 (63.1%) reported concerns related to school. Within this category, we formed several subcategories. The largest sub-category, with N = 93 (43.5%), was “stress at school”. In this category, we subsumed statements about excessive demands at school; pressure to perform; worries about lack of progress; difficulties in completing school projects and writing assignments, papers, or pre-scientific work; fear of failure in exams, tests, or presentations; fear of failing a class or school; worries about grades; problems in single subjects; school anxiety; and learning blockages.

Another N = 19 (8.9%) were worried about school organization. This category included statements about the burden of e-learning and distance learning. Respondents noted that teaching content could not be delivered as well online and that technical glitches caused delays. They felt left on their own to master the content they were learning. In addition, there were several statements in this category about the lack of rhythm due to the constantly changing organization of the school day. Alternations between face-to-face teaching and distance learning were experienced as stressful. Respondents mentioned the shift system as particularly burdensome, commenting on the chaos of school organization and the strain of interrupted routines.

N = 17 (7.9%) respondents were concerned about their upcoming school-leaving examination since this examination influenced their access to higher education institutions and universities in Austria. They expressed fears of not being able to pass the school-leaving examination. Additionally, they pointed out concerns about the postponement of the school-leaving examination and a possible devaluation due to an intended reduction in the relevant examination content. N = 6 (2.8%) also expressed being burdened by teachers who were perceived as demanding, not understanding, or unfair. Furthermore, respondents described that teachers were not competent at teaching online.


*“Question 1: What currently gives you most cause for concern?”*

*Respondent 3617: “The school, because I’m graduating in a few months. It is still unclear how exactly/if the oral Matura will take place. On top of that, due to the shift system or the resulting hybrid teaching, which is clearly worse from a school perspective than pure distance learning, we don’t really have optimal conditions to prepare for the Matura in the best possible way, as there are always technical problems, etc., which lead to slower progress in the lessons. On the other hand, I don’t know yet what I want to study after the Matura or whether studying is right. However, the worries about school/the Matura outweigh the other concerns”.*


#### 3.2.2. Restrictions (35.0%)

The second-largest category in the responses to question 1, as reported by N = 75 (35.0%), referred to the mandated restrictions due to the pandemic. This category was composed of several subcategories. N = 34 (15.9%) were troubled by the lack of social contacts, pointing out that they missed seeing friends and family members. N = 29 (13.6%) reported being burdened by the restrictions on public life, e.g., the closure of shops, restaurants, bars, clubs, fitness studios, or sports clubs, the requirement to wear a mouth-and-nose mask in public, and the lockdowns. In their statements, they described difficulties staying at home and being in the same environment all the time without any distractions. They repeatedly referred to a loss of freedom and autonomy and the feeling of being imprisoned. A further N = 5 (2.3%) particularly referred to a lack of exercise; N = 4 (1.9%) mentioned travel restrictions and N = 3 (1.4%) experienced quarantine.


*“Question 1: What currently gives you most cause for concern?”*

*Respondent 2530: “Lack of freedom. I like to live in freedom, but now I feel so dependent. Dependent on the masks, on the shops that are open, dependent on school... Every day looks the same now. I am a person who likes and needs a lot of diversion, exciting events, space for creativity, time for creativity, different sporting activities, etc. The monotony of my days is really bothering me”.*


#### 3.2.3. Self-Related Concerns (31.8%)

The third-largest category, named by N = 68 (31.8%), addressed self-related concerns. This category covered several subcategories, e.g., worries about the future, which was expressed by N = 22 (10.3%). Respondents were concerned about their own future in general, educational paths, and job opportunities. In their statements, the young people spoke of the uncertainty of the future, their fears about the future, and what life would be like for them. They also spoke of a lack of prospects, feelings of hopelessness, and their fear of making the wrong decisions. Concrete worries were also mentioned, such as being unable to find an internship or get a driver’s license.

N = 13 (6.1%) reported having negative thoughts and feelings that bothered them, e.g., feeling worthless, powerless, unable to be happy, angry, sad, or like they were disappointing others. Some more frequent mentions were coded separately. N = 8 (3.7%) mentioned a lack of drive; N = 7 (3.3%) described feeling like they were missing out on something, e.g., referring to their 18th birthday in quarantine; and N = 3 (1.4%) reported feeling lonely. Mental health stresses, such as a depressive disorder, were named by N = 6 (2.8%); N = 6 (2.8%) also reported physical health issues. Another N = 3 (1.4%) mentioned being worried about their body.


*“Question 1: What currently gives you most cause for concern?”*

*Respondent 3771: “Hopelessness, fear of the future, uncertainty. I question decisions regarding further education and job search. Often so many things weigh on me that I don’t know where to start thinking. I can no longer look forward to anything or manage to be happy, even if I am doing well or great things are happening to me. I simply lack the “positive foundation” (Maslow’s pyramid of needs describes it best)”.*


#### 3.2.4. Further Concerns

N = 27 (12.6%) mentioned relational problems with family members, their partner, and friends and peers. In addition, they reported worrying about others, mostly family members. Another category in the responses to question 1, as reported by N = 25 (11.7%), was the emergence of the pandemic. N = 12 (5.6%) gave answers that consisted exclusively of one word, e.g., “Corona”, “Pandemic”, or “Virus”. N = 13 (6.1%) referred explicitly to the uncertainty of the future: how long the pandemic will last, what else will happen, or what will happen next.

A further category was related to concerns associated with current societal developments. N = 11 (5.1%) indicated that political decisions worried them, and they saw democracy at risk. Concerns about further economic development were also expressed and worries related to the climate and environmental crisis. “Other concerns”, as reported by N = 11 (5.1%), was a catch-all category for statements that could not be assigned to any of the other categories, e.g., statements such as “private problems”. For N = 5 (2.3%), alcohol, drugs, or excessive internet use was an issue of concern. N = 7 (3.3%) said that nothing was causing them concern.

### 3.3. Greatest Current Support

In the following, we report results to question 2: What is currently providing you with the most support? Answers to this question were shorter and more bullet-pointed on average than answers to question 1; therefore, we did not include quotes. The results for question 2 are summarized in Figure 2 and are now described in more detail.

When asked what currently provides the most support, N = 153 (71.5%) mentioned social contacts, with friends mentioned most often by N = 64 (29.9%), followed by family members, named by N = 36 (16.8%), and their partner, named by N = 21 (9.8%). N = 6 (2.8%) also mentioned classmates, N = 4 (1.9%) mentioned one best friend, and N = 3 (1.4%) referred to other people more generally. Talking to someone, either in-person, over the phone, or via the internet, was found to be helpful by N = 19 (8.9%).

Recreational activities were mentioned as the most significant source of support by N = 84 (39.3%). In particular, respondents found sports activities to be helpful. N = 30 (14%) cited indoor and outdoor activities, such as jogging, cycling, horseback riding, weight training, and yoga. Going for a walk was reported by N = 9 (4.2%). In addition, N = 24 (11.2%) found listening to music helpful; N = 8 (3.7%) found it beneficial to actively relax by taking a bath, lying down, or watching relaxation videos. N = 5 (2.3%) liked to read and N = 8 (3.7%) generally referred to hobbies as a source of support.

N = 39 (18.2%) listed responses that we subsumed under the category “attitudes and abilities”. N = 30 (14.0%) mentioned mental abilities, for example, practicing self-reflection, dealing with their problems actively, thinking positively, trusting in themselves, meditating, pursuing their goals, or laughing. Creating structure for themselves, e.g., by structuring their day and making plans and to-do lists, was experienced as helpful by N = 5 (2.3%). In addition, N = 4 (1.9%) found it helpful to express their emotions, such as anger or sadness.

Distraction was experienced as the greatest source of support by N = 36 (16.8%). N = 10 (4.7%) mentioned engaging in video games alone or with friends online. Television was found to be helpful by N = 8 (3.7%); N = 6 (2.8%) liked to use social media and surfing the internet was cited by N = 5 (2.3%). Shopping was also reported as helpful by N = 2 (0.9%). N = 5 (2.3%) generally referred to “distraction” as a source of support.

We defined the mention of consuming alcoholic beverages or smoking cigarettes, as reported by N = 5 (2.3%), as a subcategory of “escape” with N = 23 (10.7%). We also subsumed the sub-categories “sleeping”, reported by N = 12 (5.6%), and “eating”, reported by N = 6 (2.8%) under “escape”.

Other sources of support included creative activities, such as writing, drawing, or making music, which was mentioned by N = 10 (4.7%); professional help from psychologists, therapists, and doctors was cited by N = 7 (3.3%), and pets were mentioned by N = 5 (2.3%). N = 3 (1.4%) found support in their faith. For N = 3 (1.4%), the school was a source of support. N = 8 (3.7%) reported that nothing was supporting them. N = 3 (1.4%) indicated they did not know what was helping. N = 3 (1.4%) stated that they did not need support.

## 4. Discussion

The present study aimed at exploring young people’s concerns and sources of support after the first year of the pandemic. A qualitative content analysis approach was used to analyze two open-ended survey questions. One of our major results was that students perceived school as their greatest current concern. They reported suffering from excessive demands and pressure to perform. Many mentioned difficulties in certain subjects and a fear of failing in a class or at school. The teachers were mostly described as not very understanding. Distance learning and the lack of rhythm in which attendance at school was organized were described as particularly stressful. The results highlighted how much young people’s psychological well-being depends on the school and learning environment. The analysis of a comparable question in a representative adult sample from Austria showed that for adults, concerns related to work were only ranked third [11]. Only 15.3% of respondents in the adult sample said they had worries related to work compared with 63.1% in our sample who had worries related to school. Obviously, for young people, staying at home and relying mainly on distance learning is very different from everyday life in classrooms and schools [31].

School-related worries were not perceived as such a burden by young people in comparable qualitative studies. Of the studies we reviewed, only Scott et al. [12] found that adolescents in the United States described academic issues as their biggest challenge. Young people from other countries did mention being burdened by schoolwork and distance learning [14,16,17], yet other concerns prevailed. In some studies, school-related worries were not mentioned at all [13,15]. These inconsistencies are in part also reflected in quantitative studies. Findings from an Australian longitudinal study by Magson et al. [32] indicated that adolescents’ greatest concern was not being able to see their friends, while they were not overly concerned with the impact that COVID-19 was having on their education. On the other hand, Ellis et al. [31] found young people from Canada to be particularly worried about schooling.

The inconclusive results from different studies may be attributed to the nature of the research questions, different methodological approaches, and different times of data collection. However, the high level of school-related concerns in our study may also be related to the specific situation in Austria. Although distance learning should have been available to teachers as an option for several years, it has not been implemented [33]. When the pandemic hit Europe, there was a lack of appropriate IT infrastructure, no ready-to-use standard learning platform, and most teachers did not have the technical and didactic skills to successfully move from classroom teaching to distance learning within a short period [33]. In June 2020, the Federal Ministry of Education, Science and Research started implementing strategies to promote digitally supported education and innovative teaching and learning formats in Austria [34]. They contributed to the rapid development of digital skills and recent research suggests that both teachers and students have become more adapted to distance learning formats and the use of digital tools over the course of the pandemic [35,36]. However, there are also indications that many students still felt overburdened by the high demands at school and by working independently on their assignments at home [37]. A majority of students missed having contact with their teachers and getting individual feedback on learning assignments [38]. Other research revealed that teachers experienced considerable stress as a result of the COVID-19 pandemic [39,40,41]. Many of them may have lacked the resources needed to best manage the difficult situation for all students. This is in line with our findings and indicates that best practices are still being explored and both teachers and students need more time to adapt and feel at ease with the new educational realities. An additional explanation for the high scoring of school-related concerns may be found in the timing of the survey, which took place in February. During wintertime in Austria, outdoor activities are limited; therefore, the school was even more important as a supportive environment.

Another relevant thematic area in our study relates to young people’s concerns about the future. Only Riiser et al. [17] also identified concerns regarding learning outcomes and prospects among their respondents; these are not addressed in the other qualitative studies reviewed for this paper. Their study, as well as ours, is one of the few that was conducted at a later point in the pandemic’s progression, namely, in December 2020, suggesting that concerns about the (educational and/or occupational) future only become apparent as the pandemic progressed and psychological distress persisted.

With regard to the sources of support, two results were particularly striking. First, with more than 10% of respondents in the category “escape”, a relatively high number of young people cited maladaptive strategies as their “main source of support”. This is concerning and underlines the need for psychosocial support services. However, second, the number of young people seeking professional help is worryingly low at 3.3%. This may be due to different reasons. Young people may not know what help is available. The shortage of psycho-social care for adolescents in Austria was already known before the pandemic. Despite years of efforts to expand psychosocial services [42], there are still gaps in the care infrastructure, which became more apparent as adolescent mental health declined during the pandemic [43]. Furthermore, there is still a stigma attached to mental illness. Feelings of shame and self-blame can prevent young people from seeking adequate help [44,45,46,47], as can the fear of being misunderstood [48].

### 4.1. Suggestions and Recommendations

Based on our findings, we would like to make several suggestions to improve the situation of young people. First, it might be beneficial to sensitize teachers to the fact that students feel most burdened by school issues and that an adjustment of requirements to the capacities of individual students is necessary. An additional option could be to increase external support for students and teachers, e.g., through social workers, counselors, psychologists, psychotherapists, or additional extracurricular teaching opportunities for students. Second, considering the high burden of distance learning on our respondents, we recommend that school closures should be considered very carefully in the future. Negative consequences might outweigh the benefits, especially since school closures can be less effective than other measures to contain the spread of SARS-CoV-2 [49]. From the statements of our respondents, it is clear that the shift system (part online, part face-to-face) brought little relief. Branquinho et al. [50] also showed that young people, especially women, were pessimistic about the new school reality in the shift system. Since contact with friends was by far the most crucial source of support for our respondents (which was in line with findings from most other studies), we argue that young people should have the opportunity to experience face-to-face social contact within the school environment in the future. Third, the high prevalence of mental health symptoms among adolescents and the reluctance of young people to seek professional help requires an expansion of low-threshold psychosocial support services. Since our study also showed that young people are most likely to confide in friends, youth-led peer support initiatives seem to be an effective low-threshold intervention for young people who are struggling with psychological problems [51,52,53,54]. Embedding youth-led peer interventions in traditional mental health services could not only be more acceptable to young people but also pave the way for more inclusive and participatory approaches that actively involve young people as agents rather than just addressees [55].

Fourth, as the pandemic had a massive impact on economic development worldwide and led to an increase in various forms of economic inequality [56], we believe it is essential for policymakers to offer young people economic prospects and provide appropriate assistance in the field of education and the labor market.

### 4.2. Limitations

This study had several limitations. For one, the answers were collected within the framework of an online survey. Although many of the answers were extensive, they still lacked the depth that can be generated in a qualitative interview. We propose further in-depth qualitative research to deepen the results of our study. Furthermore, we calculated the frequencies of our categories to illustrate how often specific topics were mentioned by respondents. However, as some answers were more detailed than others and were coded multiple times, this resulted in some answers being overrepresented in the category system. Finally, while the size of the original sample had the advantage that we were able to select a representative sample by gender and migration background for the qualitative analysis, there are other characteristics that are known to have an impact on adolescent well-being and mental health, such as the socio-economic background [57]. To avoid bias, future research should take this information into account, which was not collected in our survey.

## 5. Conclusions

This study, alongside international research, sheds light on possible causes for the worrying prevalence of mental health symptoms among young people and furthers our understanding of how young people experience their lives in times of COVID-19. The findings underline the tremendous need for low-threshold support services for young people. We propose that, in addition to acute crisis intervention, preventive services should also be significantly expanded. In view of the high burden of school and distance learning on the respondents, it also seems essential that policymakers consider the closure of schools very carefully in the future and, if necessary, give preference to other preventive measures. Efforts must focus on the needs of young people so that they are provided with some strategies to heal and cope with a stressful and potentially traumatizing situation.

## Figures and Tables

**Figure 1 healthcare-10-01334-f001:**
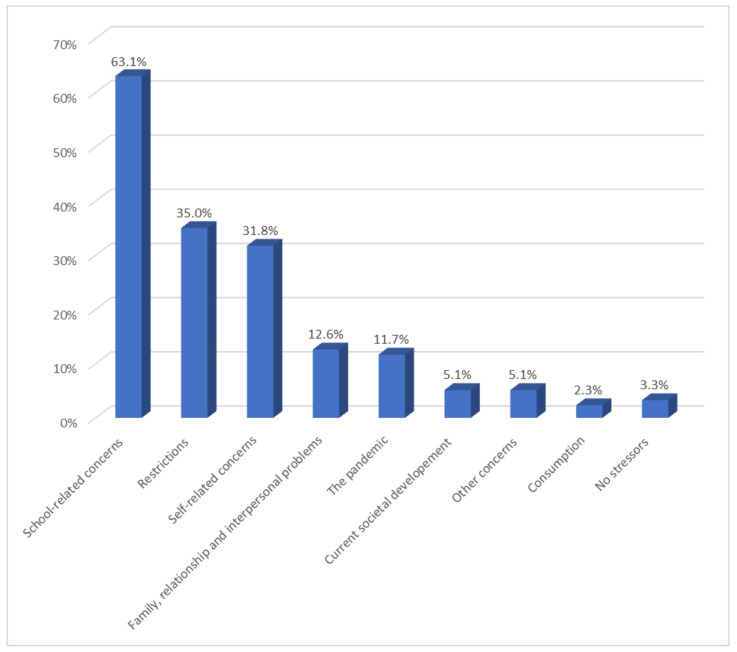
Sources of Greatest Current Concern. The percentages of participants reporting each main category of concerns that emerged from the data for question 1: “What currently gives you the most cause for concern?”

**Figure 2 healthcare-10-01334-f002:**
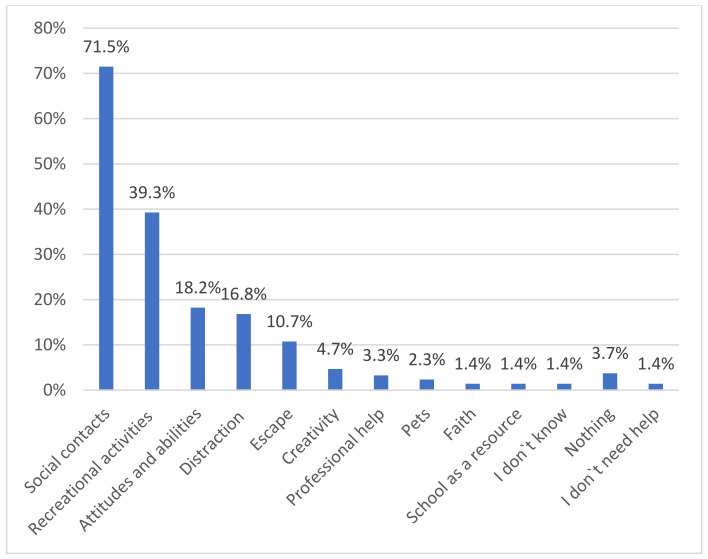
Sources of Greatest Current Support. The percentages of participants reporting each main category of support that emerged from the data for question 2: “What currently provides you with the most support?”

## Data Availability

The datasets generated during the current study are available from the corresponding author on reasonable request.

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
