# Peer review of "School Students’ Concerns and Support after One Year of COVID-19 in Austria: A Qualitative Study Using Content Analysis"

_healthcare, 2022, doi:10.3390/healthcare10071334_

Round 1
Reviewer 1 Report
The article is interesting for the reader. The main strength of the article is the content analysis approach. This is a standard qualitative research method. It is used in social science research more often. It is characterized by a certain subjectivity. However, the results are interesting. I propose that other research methods should not be mentioned. The authors do not cite results (WHO-5, PHQ-9, GAD-7, ISI, PSS-10).
Authors allow repeated phrases in conclusions, discussion. I would suggest limiting repetition of study information in conclusions.
Suggestions and recommendations can be presented separately from conclusions.
Author Response
Thank you for taking the time to offer these valuable and encouraging comments on our manuscript!
We have shortened redundancies and repetitions from the discussion and conclusion. We have also added an additional chapter on Suggestions and Recommendations.
Since there are so few qualitative studies, we find it important to review them in the introduction, even though they do not all use the same methodological approach. However, all analytical methods can be assigned to a content structuring approach.
The quantitative results (WHO-5, PHQ-9, GAD-7, ISI, PSS-10) have been published elsewhere and are referenced at the end of chapter 2.1 Research design.
Reviewer 2 Report
Adolescents suffer severely from the psychological consequences of the Covid-19 pandemic.
The authors, using qualitative content analysis, examined open-ended responses to a survey on the mental health of school students in Austria in February 2021.
A representative sample was drawn from a total survey sample of adolescents aged 14-20.
The analysis revealed several areas of concern and support. School-related concerns associated with distance learning were mentioned most frequently.
The authors concluded that: (a) compared to research conducted at the beginning of the pandemic, it appears that concerns about educational and professional futures have increased. (b) Of concern is the proportion of young people citing maladaptive coping strategies and the reluctance to seek professional support. Ideas for practice-oriented measures were developed from the study results.
The article is interesting and well written.
I have some minor comments with a pure academic spirit:
1. The abstract must better synthetize the sections of the manuscript
2. The aim is at the end of the introduction, but it is a bit hazy. Please insert a clear purpose
3. Results are arranged into a lot of themes and related paragraphs. Please introduce these themes before. Please avoid paragraphs of one or two sentences.
4. Check the resolution of the figures
Author Response
Thank you for your valuable and encouraging comments on our manuscript!
We have revised the abstract to make the breadth of the results clearer while highlighting the most important findings. We hope we have interpreted your feedback correctly.
We have added the study objectives at the end of the introduction.
In the results section, we have added an introduction to chapter 3.2. "Greatest Current Concerns", which explains the structure of the following sub-chapters. The sub-chapters correspond to the main categories of the analysis. Only the largest categories are described in separate sub-chapters. We have combined the smaller categories into one sub-chapter and thus reduced the number of sub-chapters. In addition, we have summarised short paragraphs. Since chapter 3.2. "Greatest Current Support" is considerably shorter, we have dispensed with subchapters here. We hope that this presentation is now clearer.
The figures we have integrated are in a very high resolution. Perhaps some quality is lost through the upload. We hope that a high resolution will be possible in case of publication.